# Evaluating an Early Risk Model for Uncomplicated Hypertension in Pregnancy Based on Nighttime Blood Pressure, Uric Acid, and Angiogenesis-Related Factors

**DOI:** 10.3390/ijms26136115

**Published:** 2025-06-25

**Authors:** Isabel Fernandez-Castro, Nestor Vazquez-Agra, Ana Alban-Salgado, Mariña Sanchez-Andrade, Susana Lopez-Casal, Anton Cruces-Sande, Oscar Seoane-Casqueiro, Antonio Pose-Reino, Alvaro Hermida-Ameijeiras

**Affiliations:** 1Department of Internal Medicine, University Hospital Complex of Vigo, 36312 Vigo, Pontevedra, Spain; isabelfdzcastro@gmail.com; 2Department of Internal Medicine, University Clinical Hospital of Santiago de Compostela, 15706 Santiago de Compostela, A Coruña, Spain; oscar.seoane.casqueiro@sergas.es (O.S.-C.); antonio.pose.reino@sergas.es (A.P.-R.); 3Health Research Institute of Santiago de Compostela (IDIS), 15706 Santiago de Compostela, A Coruña, Spain; anton.cruces@usc.es; 4Department of Psychiatry, Radiology, Public Health, Nursing, and Medicine, University of Santiago de Compostela (USC), 15706 Santiago de Compostela, A Coruña, Spain; 5Laboratory of Biochemistry and Clinical Analysis, University Hospital of Santiago de Compostela, 15706 Santiago de Compostela, A Coruña, Spain; ana.alban.salgado@sergas.es; 6Obstetric Service, University Hospital of Santiago de Compostela, 15706 Santiago de Compostela, A Coruña, Spain; marina.sanchez-andrade.santiso@sergas.es (M.S.-A.); susana.lopez.casal@sergas.es (S.L.-C.)

**Keywords:** gestational, pregnancy, hypertension, blood pressure, monitoring, nighttime, uric acid, angiogenesis

## Abstract

Uncomplicated hypertension (UH) during pregnancy represents a common condition, worsening maternal and fetal prognosis. However, no single biomarker has proven optimal for determining the risk of UH. We developed an early risk multivariate model for UH, integrating hemodynamics with biochemistry, focusing on the relationship between blood pressure (BP) indices, uric acid (UA), and angiogenesis-related factors (AF). We collected and analyzed data on 24 h ambulatory BP monitoring, demographic, epidemiological, clinical, and laboratory variables from 132 pregnancies. The main predictors were BP indices and serum UA and AF levels. Uncomplicated hypertension, defined as the presence of gestational hypertension or worsening of essential hypertension beyond the 20th week, was the main outcome. The combined second-degree polynomial transformation of UA and the AF (sFlt-1/PIGF) ratio, called the UA-AF Index, consistently showed a positive association with UH. The models incorporating nighttime BP indices combined with the UA-AF Index outperformed the others, with the best-performing model based on the nocturnal systolic BP (SBP). Specifically, in the best-fitting model (nighttime SBP + UA-AF Index as predictors), each 1 mmHg increase in nocturnal SBP was associated with a 10% higher risk of UH, while each one-unit increase in the UA-AF Index raised the likelihood of UH by more than twofold (accuracy: 0.830, AUC 0. 874, SE 0.032, *p*-value < 0.001, 95%CI 0.811–0.938). The combination of nighttime blood pressure indices, serum uric acid, and angiogenesis-related factors may provide added value in the assessment of uncomplicated hypertension during pregnancy.

## 1. Introduction

Uncomplicated hypertension (UH) during pregnancy represents a common condition, contributing to increased maternal and fetal morbidity and mortality. Both gestational hypertension (GH) and worsening of previously controlled hypertension (HT) beyond the 20th week of pregnancy may evolve into more severe hypertensive disorders, such as preeclampsia, particularly in women with preexisting cardiovascular risk (CVR) factors [1,2].

During a normal pregnancy, systemic vascular resistance decreases due to hormonal and endothelial changes, leading to a physiological drop in blood pressure (BP), typically in the first and second trimesters [3]. However, this adaptive response may be blunted in women with advanced maternal age, obesity, diabetes mellitus (DM), and essential HT. In these populations, elevated office and out-of-office BP levels have been associated with a higher risk of HT related complications, including intrauterine growth restriction and preterm delivery [4].

Ambulatory BP monitoring (ABPM) is superior to office BP measurement for predicting pregnancy outcomes at least for severe HT [5]. Indeed, ABPM devices recommended for use in pregnancy are more accurate than those used for office measurement or home BP monitoring [6]. However, some other maternal outcomes related to HT in pregnancy and the specific implications of individual ABPM indices remain largely uninvestigated. 

At a biochemical level, hyperuricemia in hypertensive pregnancies identifies women at increased risk of adverse maternal and fetal outcomes [7]. As the final product of purine metabolism, uric acid levels are increased in states of increased cell turnover, oxidative stress, and impaired renal clearance—all relevant to pregnancy-related hypertensive states. Elevated uric acid concentrations have been associated with endothelial dysfunction, reduced nitric oxide bioavailability, and systemic inflammation, suggesting a mechanistic role in placental and vascular injury [8]. In addition, angiogenesis-related factors —highlighting placental growth factor (PlGF), soluble fms-like tyrosine kinase-1 (sFlt-1), and human chorionic gonadotropin (hCG)—have shown utility in identifying pregnancies at risk for preeclampsia and other placental syndromes. These markers reflect trophoblast function, vascular remodeling, and placental perfusion, providing an insight into homeostasis in the functional and structural regulation of the vasculature [9,10].

Despite all this knowledge, no single biomarker has proven optimal to predict UH beyond the 20th week of pregnancy. Based on the literature reviewed, we believe that specific ABPM indices, angiogenesis-related factors, and uric acid levels measured during the first trimester of pregnancy may provide valuable predictive information for UH beyond 20 weeks of gestation. Therefore, this study aimed to develop an early risk multivariate model for UH integrating hemodynamics (24 h ABPM indices) with biochemistry (focusing on the relationship between uric acid levels and angiogenesis-related factors). 

## 2. Results

### 2.1. General and Comparative Findings: Relationship Between Blood Pressure Indices and Biochemical Markers with Uncomplicated Hypertension During Pregnancy

A total of 132 pregnant women were included, with a median age of 36 years. Among them, 55 (42%) had preexisting essential HT at the time of recruitment, whereas 77 (58%) were normotensive at baseline. Women with UH during pregnancy showed a higher prevalence of prior smoking and essential HT, as well as more frequent use of antihypertensive treatment, compared to those who remained normotensive at the 20th week. The groups were comparable in other baseline characteristics, including age and BMI. Regarding biochemical variables, patients were also comparable in fasting plasma glucose, urea, UA, and AF levels. These findings are summarized in Table 1.

According to the BP figures, patients with UH consistently exhibited higher 24 h, daytime, and nighttime BP levels, along with significantly reduced nocturnal SBP and DBP dipping compared to the control group. These results are illustrated in Figure 1 with the complete findings available in Appendix A.

### 2.2. Combined Polynomial Transformation of Uric Acid and Angiogenesis-Related Factors Ratio (UA-AF Index) as a Marker for UH

Among the models tested, the second-degree polynomial function yielded the best balance between fit and generalizability, minimizing overfitting (train/test accuracy = 0.70/0.67). Therefore, the UA-AF Index corresponded to the degree two polynomial transformation. The results related to the polynomial transformations for UH are summarized in Table 2, and the performance metrics of the models are illustrated in Figure 2.

### 2.3. Identifying the Optimal Model Based on Blood Pressure Indices and the UA-AF Index for the Risk of UH

Logistic regression analysis was performed on each BP index, adjusting for potential confounders identified in the previous analyses. The main findings were as follows: (1) all BP indices were positively associated with the risk of UH; (2) in all logistic regression models, the UA-AF Index consistently showed a positive association with UH; (3) those models incorporating nighttime BP indices combined with the UA-AF Index outperformed the others, with the best-performing model based on the nocturnal SBP. Specifically, in the best-fitting model (nighttime SBP + UA-AF Index as predictors), each 1 mmHg increase in nocturnal SBP was associated with a 10% higher risk of UH, while each one-unit increase in the UA-AF Index increased the likelihood of UH by more than twofold after the 20th week (accuracy: 0.83, AUC 0.874, SE 0.032, *p*-value < 0.001, 95%CI 0.811–0.938). Table 3 summarizes the logistic regression models based on the office and nighttime BP indices, both with and without considering the UA-AF Index. The ROC curves depicting model performance are shown in Figure 3. Results from models using other 24 h ABPM indices are presented in the Appendix A.

Bayesian hypothesis testing on the model accuracy yielded the following insights: (1) the posterior probability that the best model for SBP (based on nighttime SBP + UA-AF Index) outperformed the baseline model based on office SBP was 0.997, with a 95% credible interval for the difference that excludes zero, and (2) the posterior probability that the best model for DBP (nighttime DBP + UA-AF Index) was superior to the baseline model based on office DBP was 0.988, also with a credible interval that does not include zero. All results are shown in Figure 4.

## 3. Discussion

The findings can be summarized as follows: (1) ABPM indices—particularly the nighttime BP ones—showed a stronger association for UH than conventional BP indices; (2) the inclusion of biochemical markers, specifically uric acid and the angiogenesis-related factors ratio (UA-AF Index), improved the performance of the BP-based models for UH.

We present one of the first studies to address a frequent and clinically relevant issue in real-world practice: the high prevalence of UH developing after the 20th week of gestation, including both GH and the worsening of previously well-controlled essential HT [11]. Elevated BP beyond the 20th week has been broadly associated with adverse maternal and fetal outcomes. Although the prognostic severity is notably lower compared to other hypertensive disorders of pregnancy, the high prevalence of elevated BP at mid-gestation underscores the importance of early detection and management [12,13].

Current evidence support that ABPM is at least non-inferior to office BP measurements for diagnosing HT during pregnancy [14,15,16]. However, this study provides an individualized insight of each BP index performance, rather than a purely diagnostic perspective based on ABPM, by assessing the UH risk at the 20th week. 

Nocturnal hypertension has been investigated as a potential predictor of preeclampsia and other severe hypertensive disorders of pregnancy [17,18,19]. Elevated nocturnal BP during pregnancy, even in the absence of preeclampsia criteria, may reflect early cardiovascular maladaptation [20]. It has been associated with a loss of the normal dipping pattern, increased peripheral vascular resistance, subclinical endothelial dysfunction, and an altered response of the renin–angiotensin system [21,22,23]. Additionally, a low-grade chronic inflammatory state may contribute to sustained vasoconstriction during rest [24]. In this context, higher nighttime BP levels may also serve as an early warning signal for UH development, enabling closer monitoring in selected subgroups and enhancing preventive care strategies.

Experimental studies provide further support for these clinical insights. Animal models with disrupted circadian clock genes—such as Bmal1 or Per2—exhibit impaired nocturnal BP regulation, heightened sympathetic tone, and increased vascular stiffness, all of which mimic features observed in gestational hypertension [25,26]. In parallel, in vitro studies using endothelial cells have shown that circadian misalignment reduces endothelial nitric oxide synthase (eNOS) expression and enhances oxidative stress via NADPH oxidase activation, compromising vascular relaxation during rest periods [27,28]. These findings suggest that even subtle alterations in circadian regulation at the molecular or cellular level may translate into early hemodynamic changes detectable through nighttime ABPM. In the context of pregnancy, where systemic vascular adaptation and placental remodeling are tightly regulated, such dysregulation may contribute to the development of UH in women who appear clinically normotensive in early gestation.

As previously introduced, hypertensive states in pregnancy reflect pathogenic pathways and a set of physiopathological processes that lead to a dysfunctional internal milieu [29]. This allows for the measurement of several biochemical markers involved in these processes and associated with BP indices [30,31]. To date, serum uric acid levels and angiogenesis-related factors have primarily been linked to complicated hypertensive disorders of pregnancy [32,33,34]. However, the results might support the potential utility of combining these biomarkers (into the UA-AF Index) for assessing the risk of UH beyond the 20th week of gestation, representing a key biochemical dimension.

These findings are consistent with basic and animal studies implicating endothelial dysfunction and altered vascular tone regulation [35]. Uric acid levels are closely linked to oxidative stress imbalance, involving increased NADPH oxidase activity and disruption of nitric oxide signaling pathways [36,37]. Furthermore, trophoblastic factors exert a major influence on angiogenic balance during pregnancy. Notably, sFlt-1 antagonizes not only PlGF but also vascular endothelial growth factor (VEGF), which plays an essential role in angiogenic regulation beyond pregnancy [38,39,40]. This highlights the importance of early biomarker and ABPM evaluation to prevent hypertensive disorders in pregnancy.

In summary, this study represents a preliminary multidimensional approach to uncomplicated hypertension developing after the 20th week of pregnancy, integrating both hemodynamic (ABPM-derived indices) and biochemical (uric acid and angiogenesis factors) factors, and showing a good performance within this patient cohort. Given that patients with gestational hypertension or worsening blood pressure levels during pregnancy are at increased risk of developing more severe hypertensive disorders, early identification could enable greater anticipation and expand the capacity for timely intervention, thereby adding value in minimizing adverse maternal and fetal outcomes.

These findings underscore the need for future validation and cost-effectiveness studies, with the ultimate goal of integrating these results as a predictive tool into routine clinical practice and guidelines, in order to more accurately anticipate uncomplicated hypertension during pregnancy and reduce associated complications.

### Limitations and Strengths

This was a preliminary modeling and real-world clinical practice study with limitations inherent to its observational design and consecutive sampling approach. The pregnancies that underwent ABPM may differ in baseline characteristics compared to other pregnant populations, such as having a higher prevalence of preexisting HT. This could limit the generalizability of the results to other populations. Moreover, the variables were collected at a single time point, which does not account for their intrinsic variability over time. Regarding the UA-AF index, performance metrics were estimated exclusively from cross-validation to prevent optimistic bias. The model equation for the UA-AF Index, trained on the full dataset, was intended for practical implementation, not performance reporting. Since the final multivariate models for BP indices and the UA-AF Index were fitted on the full dataset without cross-validation, reported performance metrics may be subject to optimistic bias. Thus, further external validation of these findings in independent and larger cohorts will be necessary to confirm the generalizability and clinical utility of the proposed model. Finally, the Bayesian comparison provides a probabilistic interpretation of model superiority, avoiding reliance on *p*-values. Instead of a traditional Bayes Factor, we report the posterior probability and credible interval for the difference in accuracy, which directly reflects the strength and uncertainty of the observed improvement.

## 4. Methods and Materials

### 4.1. Study Design, Setting, and Participants

This was a prospective observational study conducted in the Department of Internal Medicine at the University Hospital of Santiago de Compostela during 2024. Regarding inclusion criteria, we consecutively enrolled the following: (1) pregnant women aged 18 years or older during the first trimester of pregnancy (up to 12 weeks of gestation); (2) with no prior history of HT or with well-controlled essential HT; (3) presenting at least a moderate CVR, as defined by major clinical practice guidelines; and (4) referred from the obstetrics department for comprehensive CVR assessment and 24 h ABPM in accordance with our internal protocols for comprehensive CVR evaluation in pregnant patients [41]. The exclusion criteria were current smoking (within 6 months prior to pregnancy), any alcohol consumption, and established cardiovascular disease. The presence of contraindications to ABPM, whether psychological–behavioral, sleep-related, or anatomical–physiological, was also considered an exclusion criterion [41,42].

These patients were prospectively followed to assess the development of UH, as defined in the corresponding section below. The diagnosis of UH was confirmed based on office BP readings obtained after the 20th gestational week, either at our unit or by the referring obstetrics team. Then, we assessed the relationship between UH after the 20th week of pregnancy and a set of independent variables, including clinical–anthropometric features, biochemical parameters, and ABPM indices measured within the first weeks of pregnancy. 

### 4.2. Clinical and Laboratory Variables

We collected data on participants’ age, sex, and history of tobacco use (categorized as no/yes). Body mass index (BMI) was calculated as weight divided by height squared (kg/m^2^) [43]. Office BP was measured in accordance with the STRIDE BP protocol, using the WatchBP Office oscillometric device (Microlife Corporation, Widnau, Switzerland) [44]. Blood samples were obtained at 08:00 AM after a 12 h overnight fast, on the same day as office BP measurements and 24 h ABPM implementation. Serum uric acid (UA) levels were quantified using colorimetry with the Atellica Solution system (Siemens Healthcare Diagnostics, Tarrytown, NY, USA). The angiogenesis-related factors (AF)—placental growth factor (PlGF) and soluble fms-like tyrosine kinase-1 (sFlt-1)—were measured using electrochemiluminescence immunoassay (ECLIA) techniques with the Cobas e411 system (Roche diagnostics, Zug, Switzerland). The AF ratio was computed as the relation between sFlt-1 and PIGF (sFlt-1/PIGF ratio).

### 4.3. Parameters of 24-h ABPM Collection

We followed the methodology used in our previous studies, consistently adhering to the recommendations outlined in major consensus documents [41,44,45,46]. Patients underwent 24 h ABPM in compliance with STRIDE BP standards, using one of the following validated oscillometric devices: Space-Labs 90207 (Space-Labs Inc., Redmond, WA, USA), Microlife WatchBP O3 (Microlife Corporation, Widnau, Switzerland), and Cardioline Walk 200b (AB Medica Group, S.A., Barcelona, Spain) [44]. Blood pressure readings were taken every 20 min during the daytime and every 30 min at night, with time periods determined based on patients’ self-reports. The test was considered reliable if more than 70% of the expected measurements were valid. On the day of monitoring, patients completed a form detailing their sleep times, medication use, and any issues experienced during the recording process. When the patient explicitly reported disrupted sleep during the test, the 24 h ABPM was repeated. 

The following average indices were obtained: 24 h SBP (24-hSBP), daytime SBP (dSBP), and nighttime SBP (nSBP); 24 h DBP (24-hDBP), daytime DBP (dDBP), and nighttime DBP (nDBP). As for the relationship between daytime and nighttime BP levels, nSBP dipping and nDBP dipping were calculated as the ratio of (a) the difference between daytime and nighttime indices to (b) the daytime index, expressed as a percentage [47,48].

### 4.4. Uncomplicated Hypertension During Pregnancy

This variable was considered the presence of GH or worsening of essential HT beyond the 20th week, given the prognostic implications of two highly prevalent and under-investigated maternal outcomes. Gestational hypertension was defined as office systolic and diastolic BP (SBP, DBP) values equal to or greater than 140 mmHg and/or 90 mmHg, respectively, in accordance with the clinical practice guidelines of the European Society of Hypertension (ESH). Worsening of preexisting HT was defined as a decompensation of chronic HT reaching the same BP thresholds and moment as GH, in order to standardize classification criteria across the study sample [41].

### 4.5. Ethical Statement

This study was conducted in accordance with the ethical principles outlined in the Declaration of Helsinki and the standards of good practice (NBP) in research. Before participation, all patients received comprehensive information about the study to ensure their understanding and voluntary consent, and informed consent was obtained. The study protocol was formally approved by the Research Ethics Committee of Santiago-Lugo, underscoring our commitment to ethical standards in biomedical research (code 2022/144).

### 4.6. Statistical Analysis

Statistical analysis was performed using SPSS 22.0 (SPSS Inc., Chicago, IL, USA). Descriptive analyses were conducted, with qualitative variables expressed as the number (percentage) and quantitative variables as the median (interquartile range). For the univariate analysis, qualitative and quantitative variables were compared using the chi-squared test and the Mann–Whitney U test, respectively [49]. 

Regarding the biochemical variables, if we did not find a first-order relationship between the main predictors (UA and the AF ratio) with UH beyond the 20th week of pregnancy, a combined polynomial transformation of UA and the AF ratio (The UA-AF Index) was implemented in order to optimize the performance of both variables for UH. Logistic regression was used as the modeling approach and all performance metrics were estimated using k-fold cross-validation to ensure an unbiased evaluation of the model. The optimal transformation was selected based on its ability to improve the model performance while maintaining low complexity. We then generated and stored the out-of-fold logit values from the model, which served as the new combined variable (the UA-AF Index). Specific open-access libraries of Python 3 were used through Anaconda [50].

Multivariate analysis was performed using binary logistic regression according to UH (no/yes). Predictor variables were the BP indices and the UA-AF Index. Before constructing the final multivariate model for each BP index, preliminary logistic regression models were used to assess the potential relationships of BP indices and UA-AF Index with covariates. Interaction terms were considered relevant if their coefficients were statistically significant. Confounding variables were identified as those that altered the odds ratio by 10% or more. The selection of the optimal model for each BP index involved a thorough evaluation, including the omnibus test of model coefficients for initial validation, the Hosmer–Lemeshow test for goodness of fit, and adherence to the assumptions of binary logistic regression. In the final models, only coefficients with statistical significance (*p* < 0.05) were deemed relevant [51].

The performance of the models based on BP indices and the UA-AF Index was evaluated using ROC curves. Hypothesis testing was conducted using Bayesian analysis, where the alternative hypothesis (H_1_) stated that a model of interest would demonstrate greater accuracy compared to a baseline model (null hypothesis, H_0_). A uniform Beta prior distribution, Beta(1,1), was assumed for the accuracy of each model. The posterior distributions were obtained using the conjugate Beta-Binomial update based on the observed number of correct predictions out of the total sample. We drew 100,000 samples from each posterior distribution to estimate the probability that the model of interest outperformed the baseline model. The posterior probability P(θ_1_ > θ_0_) was used to quantify the evidence in favor of H_1_. Additionally, 95% credible intervals were computed for both models’ accuracies and for the difference θ_1_−θ_0_ [52,53].

## Figures and Tables

**Figure 1 ijms-26-06115-f001:**
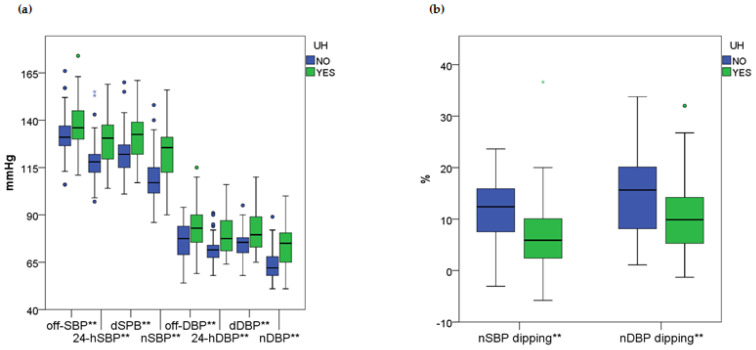
Comparison of blood pressure indices between patients with uncomplicated hypertension: no (blue) and yes (green). (**a**) Office and out-of-office blood pressure indices; (**b**) nighttime blood pressure dipping. Results marked with ** reached a *p*-value of less than 0.05. The numerical values of all indices are shown in Appendix A. UH—uncomplicated hypertension; BP—blood pressure; SBP—systolic BP; off-SBP—office SBP; 24-hSBP—24 h SBP; dSBP—daytime SBP; nSBP—nighttime SBP; DBP—diastolic BP; off-BDP—office DBP; 24-hDBP—24 h DBP; dDBP—daytime DBP; nDBP—nighttime DBP; mmHg—millimeter of mercury; %—percentage.

**Figure 2 ijms-26-06115-f002:**
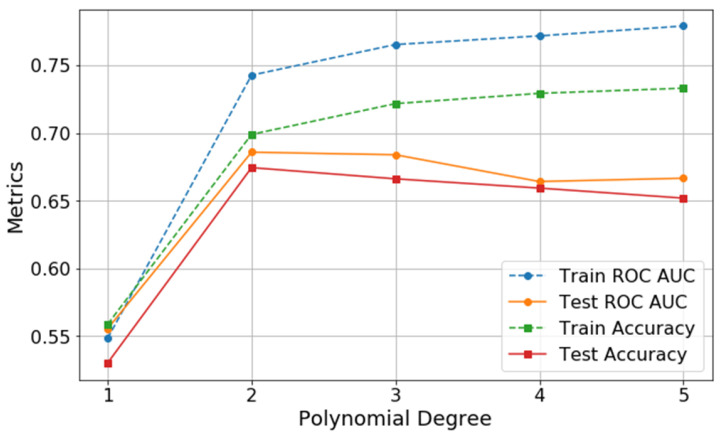
Visual representation of the combined polynomial transformations based on uric acid (UA) and the angiogenesis-related factors (AF) ratio (the UA-AF Index) for UH according to AUC and accuracy metrics. All performance metrics (Accuracy, AUC, and Log loss) correspond to the mean out-of-fold results obtained through fivefold stratified cross-validation. Note that the model based on the second-order polynomial transformation achieved the closest match between train and test accuracy. AUC—area under the curve; UH—uncomplicated hypertension; UA-AF Index—uric acid and angiogenesis-related factors ratio index.

**Figure 3 ijms-26-06115-f003:**
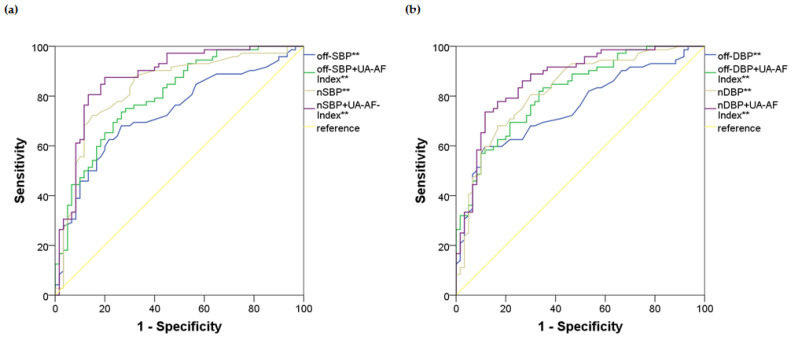
ROC curves for the logistic regression models based on blood pressure indices and the UA-AF Index (Table 3) for predicting UH. (**a**) Office and nighttime SBP indices: Off-SBP: AUC 0.731, SE 0.044, *p*-value < 0.001, 95%CI 0.644–0.817; off-SBP + UA-AF index: AUC 0.800, SE 0.038, *p*-value < 0.001 95%CI 0.725–0.875; nSBP: AUC 0.831, SE 0.037, *p*-value < 0.001, 95%CI 0.757–0.904; nSBP + UA-AF index: AUC 0.874, SE 0.032, *p*-value < 0.001, 95%CI 0.811–0.938; (**b**) office and nighttime DBP indices: Off-DBP: AUC 0.752, SE 0.042, *p*-value < 0.001, 95%CI 0.670–0.835; off-DBP + UA-AF index: AUC 0.818, SE 0.036, *p*-value < 0.001, 95%CI 0.747–0.888; nDBP: AUC 0.824, SE 0.037, *p*-value < 0.001, 95%CI 0.753–0.896; nDBP + UA-AF index: AUC 0.866, SE 0.032, *p*-value < 0.001, 95%CI 0.804–0.929. Results marked with ** reached a *p*-value of less than 0.05. ROC—receiver operating characteristics; UA-AF Index—uric acid and angiogenesis-related factors ratio index; UH—uncontrolled hypertension. AUC—area under the curve; SE—standard error; CI—confidence interval; BP—blood pressure; SBP—systolic blood pressure; off-SBP—office SBP; nSBP—nighttime SBP; DBP—diastolic blood pressure; off-DBP—office DBP; nDBP—nighttime DBP.

**Figure 4 ijms-26-06115-f004:**
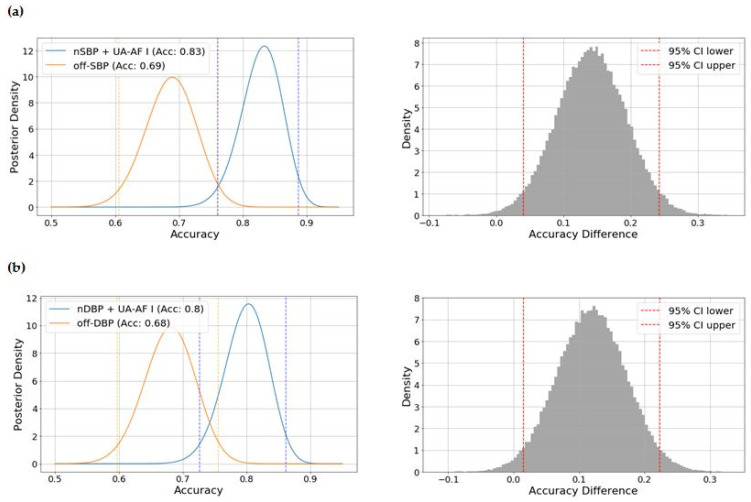
Bayesian inference on the models’ accuracy comparing (**a**) nighttime SBP + UA-AF Index (accuracy 0.83, 95% credible interval [0.760–0.887]) versus office SBP (Accuracy 0.69, 95% credible interval [0.605–0.762]), posterior probability that Model 1 > Model 2: 0.997, 95% credible interval for the accuracy difference [0.040–0.242]; (**b**) nighttime DBP + UA-AF Index (accuracy 0.80, 95% credible interval [0.726–0.861]) versus office SBP (accuracy 0.68, 95% credible interval [0.597–0.755]), posterior probability that Model 1 > Model 2: 0.988, 95% credible interval for the accuracy difference [0.014–0.223]; SBP—systolic blood pressure; DBP—diastolic blood pressure; Off-SBP—office SBP; Off-DBP—office DBP; nSBP—nighttime SBP; nDBP—nighttime DBP; UA-AF = uric acid and angiogenesis-related factors ratio index.

**Table 1 ijms-26-06115-t001:** General findings and comparisons between the groups related to uncomplicated hypertension during pregnancy.

Variable	Total (n = 132)	UH ^a^ (No) (n = 60)	UH (Yes) (n = 72)	*p*-Value
Age (years) †	36 (8.0)	35.5 (9.8)	37.0 (8.0)	0.391
Primipara (yes) ‡	49 (37.1)	21 (35.0)	28 (38.9)	0.719
N° of births (>1) ‡	15 (11.4)	6 (10.0)	9 (12.5)	0.785
Miscarriages (>1) ‡	5 (3.8)	2 (3.3)	3 (4.2)	0.999
Tobacco (yes) ^b^ ‡	16 (12.1)	3 (5.0)	13 (18.1)	0.031
CKD (yes) ^c^ ‡	6 (4.5)	2 (3.3)	4 (5.6)	0.688
DM (yes) ‡	9 (6.8)	2 (3.3)	7 (9.7)	0.181
Essential HT (yes) ‡	55 (41.7)	19 (31.7)	36 (50.0)	0.036
HT drugs (yes) ‡	22 (16.7)	4 (6.7)	18 (25.0)	0.005
BMI (kg/m^2^) †	30.8 (11.3)	31.4 (13.12)	30.4 (10.6)	0.500
Office SBP (mmHg) †	134.5 (12.8)	131.0 (10.8)	136.0 (15.0)	0.014
Office DBP (mmHg) †	81.0 (13.8)	77.5 (15.0)	83.0 (14.8)	0.002
Office HR (bpm) †	90.0 (18.0)	89.5 (22.8)	90.5 (16.8)	0.557
FPG (mg/dL) †	78.0 (13.3)	78.5 (11.8)	78.0 (17.8)	0.634
Urea (mg/dL) †	21.0 (7.0)	21.0 (7.0)	21.0 (7.0)	0.864
Uric acid (mg/dL) †	4.1 (1.38)	4.1 (1.90)	4.2 (1.08)	0.460
sFlt-1 (U/L) †	1908.0 (2683.3)	1789.0 (2869.0)	2045.5 (2302.8)	0.810
PlGF (U/L) †	215.5 (250.8)	226.5 (274.3)	201.5 (232.5)	0.805
sFlt-1/PlGF †	7.0 (22.0)	8.0 (26.0)	7.0 (19.0)	0.544

^a^ Patient groups according to the presence of uncomplicated hypertension at the 20th gestational week; ^b^ former smokers for more than 6 months at the time of recruitment; ^c^ chronic kidney disease with an estimated glomerular filtration rate lower than 60 mL/min (CKD-EPI). Results expressed as † refer to the median and interquartile range, and results expressed as ‡ refer to the number and percentage. UH—uncomplicated hypertension; CKD—chronic kidney disease; DM—diabetes mellitus; HT—hypertension; BMI—body mass index; SBP—systolic blood pressure; DBP—diastolic blood pressure; HR—heart rate; FPG—fasting plasma glucose; sFlt-1—Fms-like tyrosine kinase-1; PlGF—placental growth factor; Kg—kilogram; m—meter; dL—deciliter; mg—milligram; %—percentage.

**Table 2 ijms-26-06115-t002:** Combined polynomial transformations of uric acid and angiogenesis-related factors ratio as possible markers for uncomplicated hypertension during pregnancy.

	Accuracy		AUC		Log Loss	
Transformation Degree	*Train Set*	*Test Set*	*Train Set*	*Test Set*	*Train Set*	*Test Set*
1	0.56	0.53	0.55	0.56	0.68	0.69
2 ^a^	0.70	0.67	0.74	0.69	0.57	0.62
3	0.72	0.67	0.77	0.68	0.55	0.62
4	0.73	0.66	0.77	0.66	0.54	0.80
5	0.73	0.65	0.78	0.67	0.53	1.05

Logistic regression models were trained using polynomial transformations (degrees 1 to 5) of the UA–AF index (combining uric acid and angiogenesis-related factor ratio) to predict uncomplicated hypertension during pregnancy. All performance metrics (Accuracy, AUC, and Log loss) correspond to the mean out-of-fold results obtained through fivefold stratified cross-validation. ^a^ The degree-2 transformation showed the best performance according to model complexity and overfitting possibility. The final fitted equation corresponds to the model trained on the entire dataset for interpretability purposes: UA-AF index = 0.3855 + 0.4989 ×UA − 0.2740 × AF − 0.9713 × UA^2^ + 1.4606 × UA × AF − 0.3904 × AF^2^. UA—uric acid; AF—angiogenesis-related factors; UA-AF Index—uric acid and angiogenesis-related factors ratio index; AUC—area under the curve.

**Table 3 ijms-26-06115-t003:** Logistic regression models based on 24 h ABPM indices and the UA-AF index for predicting UH during pregnancy.

	Variable	B	SE	Wald	*p*-Value	Exp(B)	95%CI (Lower)	95%CI (Upper)
Office SBP (accuracy: 69%. Nagelkerke R^2^: 0.173)								
	Former smoker (yes)	1.678	0.685	5.997	0.014	5.354	1.398	20.507
	HT drugs (yes)	1.790	0.607	8.682	0.003	5.987	1.821	19.686
	Office SBP (mmHg)	0.043	0.017	6.348	0.012	1.044	1.010	1.079
	Constant	−6.032	2.312	6.805	0.009			
Office SBP + UA-AF index (accuracy: 74%. Nagelkerke R^2^: 0.371)								
	Former smoker (yes)	1.964	0.759	6.693	0.010	7.128	1.610	31.561
	HT drugs (yes)	1.989	0.676	8.667	0.003	7.311	1.944	27.488
	Office SBP (mmHg)	0.046	0.018	6.341	0.012	1.047	1.010	1.086
	UA-AF index (SU)	1.029	0.293	12.308	<0.001	2.799	1.575	4.973
	Constant	−6.735	2.515	7.172	0.007			
Nighttime SBP (accuracy: 76%. Nagelkerke R^2^: 0.372)								
	HT drugs (yes)	1.555	0.661	5.539	0.019	4.736	1.297	17.292
	nSBP (mmHg)	0.087	0.018	23.925	<0.001	1.091	1.054	1.130
	Constant	−10.116	2.061	24.091	<0.001			
Nighttime SBP + UA-AF index (accuracy: 83%. Nagelkerke R^2^: 0.505)								
	HT drugs (yes)	1.264	0.662	3.650	0.056	3.540	0.968	12.952
	nSBP (mmHg)	0.098	0.020	23.041	<0.001	1.103	1.060	1.149
	UA-AF index (SU)	1.039	0.327	10.113	0.001	2.826	1.490	5.361
	Constant	−11.573	2.382	23.604	<0.001			
Office DBP (accuracy: 68%. Nagelkerke R^2^: 0.264)								
	Former smoker (yes)	1.811	0.705	6.604	0.010	6.115	1.537	24.335
	HT drugs (yes)	1.764	0.608	8.423	0.004	5.838	1.773	19.22
	Office DBP (mmHg)	0.063	0.019	11.08	0.001	1.065	1.026	1.106
	Constant	−5.329	1.55	11.815	0.001			
Office DBP + UA-AF index (accuracy: 73%. Nagelkerke R^2^: 0.411)								
	Former smoker (yes)	2.081	0.782	7.076	0.008	8.010	1.729	37.110
	HT drugs (yes)	1.941	0.681	8.127	0.004	6.967	1.834	26.468
	Office DBP (mmHg)	0.067	0.020	10.710	0.001	1.069	1.027	1.113
	UA-AF index (SU)	0.998	0.286	12.145	<0.001	2.713	1.548	4.757
	Constant	−5.807	1.664	12.181	<0.001			
Nighttime DBP (accuracy: 74%. Nagelkerke R^2^: 0.380)								
	Former smoker (yes)	1.535	0.731	4.357	0.037	4.641	1.098	19.617
	HT drugs (yes)	1.653	0.649	6.492	0.011	5.223	1.464	18.625
	nDBP (mmHg)	0.117	0.026	20.861	<0.001	1.124	1.069	1.181
	Constant	−8.145	1.743	21.839	<0.001			
Nighttime DBP + UA-AF index (accuracy: 80%. Nagelkerke R^2^: 0.505)								
	Former smoker (yes)	1.828	0.876	4.352	0.037	6.221	1.117	34.653
	HT drugs (yes)	1.663	0.690	5.801	0.016	5.274	1.363	20.406
	nDBP (mmHg)	0.120	0.028	18.802	<0.001	1.128	1.068	1.191
	UA-AF index (SU)	1.004	0.327	9.408	0.002	2.729	1.437	5.182
	Constant	−8.569	1.889	20.570	<0.001			

Logistic regression models elucidating the relationship of BP indices and UA-AF index (predictor variables) with UH (no/yes) are presented. All models incorporated data from 132 pregnant women ensuring no missing data. The variables to be controlled were age, BMI, HT drugs, essential HT, and former smoking. For all the models: omnibus test of model coefficients (*p*-value) < 0.05; Hosmer–Lemeshow test (*p*-value) > 0.05. All comprehensive models including those based on the 24 h and daytime BP indices are detailed in Appendix A. 24-hABPM―twenty-four-hour ambulatory blood pressure monitoring; UH―uncomplicated hypertension; HT―hypertension; UA―uric acid; AF―angiogenesis-related factors; UA-AF Index—uric acid and angiogenesis-related factors ratio index; BP―blood pressure; SBP―systolic BP; nSBP―nighttime SBP; DBP―diastolic BP; nDBP―nighttime DBP; mmHg―millimeter of mercury; %―percentage; SU―standard unit.

## Data Availability

Data are available on reasonable request from the corresponding author. In accordance with Article 18.4 of the Spanish Constitution and the Organic Law on Data Protection and Guarantee of Digital Rights (LOPDGDD) of 6 December 2018, the privacy and integrity of the individual will be protected at all times; so, anonymous data are available upon reasonable request.

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
