# Peer review of "Evaluating an Early Risk Model for Uncomplicated Hypertension in Pregnancy Based on Nighttime Blood Pressure, Uric Acid, and Angiogenesis-Related Factors"

_ijms, 2025, doi:10.3390/ijms26136115_

Round 1
Reviewer 1 Report
Comments and Suggestions for Authors
Dear editor,
The manuscript aims to develop a multivariate model for early risk of UH that integrates hemodynamics (24-hour ABPM index), with a focus on the relationship between uric acid levels and angiogenesis related factors. However, the paper currently lacks in-depth analysis of key data, and the discussion section of the entire paper is minimal. Therefore, it is recommended that the author increase corresponding pathological research and mechanism research, and must have persuasive data support and corresponding data analysis.Therefore, the current version of the paper is not recommended for publication in this journal.
Author Response
Dear Reviewer,
We kindly invite you to review the attached document containing our detailed responses to your comments and suggestions.
We sincerely appreciate your time and consideration.
With respectful regards,
The authors

Reviewer 2 Report
Comments and Suggestions for Authors
I find this paper very confusing. What is the patient group and how were they selected? they discuss 132 who were included and all became hypertensive either worsening pre-existing HT or GH. They then say that 55 were in the first group and 82 in there second which makes 137 patients. The samples were taken "in the first weeks of pregnancy" but not specified.
So what is the initial study group? Was this from a large group who were tested and the 132 patients retrospectively selected because of development of UH? But who was no UH patients?
There were 60 patients with UH and 72 without UH but 19 of them were on HT drugs. Did that affect the results?
Was this a retrospective analysis? if this was prospective, what were the results of the majority of patients who did not develop hypertension at all.
There needs to be clearer description of how the patients were selected, and how the study was carried out as I do not find it clear. What was the hypothesis being tested.
The study appears to show that Blood pressure levels, particularly the lack of the nighttime dip is associated with UH. Ambulatory BP assessment is invasive so again how were these patients selected? This would not be acceptable to most women.
Similarly, the UA-AF Index was predictive. The heading in table 2 are out of line.
They them add several more layers of complicated statistics to show the same results and prove validity. However, this is a small sample size and to runs testing and trialling on this small data set can give false results. More research as the authors state is required.
In the discussion they discuss other papers that have found similar results and particularly discuss the value of ABPM compared to office values.
These results have limited value in their current form as it does not appear to add much to the existing algorithms.
There may well be good data here but without the basic understanding of the reason for the study, and how it was carried out it is difficult to assess fairly.
How can anyone try and emulate it is it is not possible to understand wjhat was done in this study.
Author Response

(The authors gave the same response as above.)

Round 2
Reviewer 1 Report
Comments and Suggestions for Authors
The author has basically solved all the problems I mentioned first, so it is recommended that the manuscript be published in the current version in this journal
Author Response
Dear reviewer 1,
We would like to sincerely thank you for your role in guiding the peer review process.
Kind regards,
The authors
Reviewer 2 Report
Comments and Suggestions for Authors
This version of the paper does not really improve on the first version, as I still have no understanding of how this study was carried out, what its purpose is, and what the outcomes are and the value to the scientific community.
First of all the structure of the paper itself: it's important that the methods section comes after introduction and before the results so that the readers can understand exactly what was done.
The problem is that the description of the methodology is poor. I still have no understanding of how these patients were recruited. They appear to say they picked on this number of patients who were in a monitoring programme because of risks of developing hypotension but they had no history of hypotension in the past. How could all the patients recruited develop hypotension in some way? This seems extremely illogical and unlikely.
The further thing is there is still no definition of what uncomplicated hypoertension and gestational hypertension is and how it differs from gestational hypertension. Presumably it's both new onset hypertension after 20 weeks and without complications, but what's the difference between uncomplicated hypertension and gestational hypertension? And why do we want to differentiate between the two?
There is no associated pathology of note or development of preeclampsia. If you quote things like growth restriction as an outcome of hypotension why is that uncomplicated? As that should be a complication of hypotension and therefore it makes no sense to me.
The investigations that were done were highly competently achieved but the time and invasion to the individual was quite excessive, particularly for 24hr blood pressure recordings. I'm not sure again how this could be used in day-to-day practise. Again it depends on who and what the recruitment criteria are.
The ability to differentiate between uncomplicated hypertension and gestational hypertension is of no value if it's not understood what these actually are and what benefit this would give. Does it give you information about intervention, increase monitoring, what is the benefit? What is the outcome? And I'm not sure how translatable this is to any other population.
I find it frustrating as there's a lot of good work being done in this project without adequate description of why it was done and how it was done. This makes it of little value in its current form
Comments on the Quality of English LanguageThe quality of English is adequate apart from failing to describe the methodology properly.
Author Response
Dear Reviewer 2,
We kindly invite you to review the attached document containing our detailed responses to your comments and suggestions.
We sincerely appreciate your time and consideration.
With respectful regards,
The authors

Round 3
Reviewer 2 Report
Comments and Suggestions for Authors
I thanks the authors for their efforts in answering my queries which must have been frustrating to them but with over 40 years experience in publishing in this area and developing guidelines, these clarities are important.
I feel that they describe more fully what they have done and their intentions. Also they have described there conclusion more fully.
I am still not convinced about the definition of UC separate from GH but I follow their argument.